# Grief and Bereavement in Parents After the Death of a Child in Low- and Middle-Income Countries

**DOI:** 10.3390/children7050039

**Published:** 2020-05-01

**Authors:** Michael J. McNeil, Eve Namisango, Jennifer Hunt, Richard A. Powell, Justin N. Baker

**Affiliations:** 1Department of Hospice and Palliative Medicine, University of Tennessee Health Science Center, Memphis, TN 38103, USA; 2Department of Oncology, Division of Quality of Life and Palliative Care, St. Jude Children’s Research Hospital, Memphis, TN 38105, USA; justin.baker@stjude.org; 3African Palliative Care Association, P.O. Box 72518, Kampala, Uganda; eve.namisango@africanpalliativecare.org; 4Cicely Saunders Institute of Palliative Care, Policy & Rehabilitation, King’s College London, London WC2R 2LS, UK; 5Avondale, P.O. Box A222, Harare, Zimbabwe; jhunt@mango.zw; 6MWAPO Health Development Group, Nairobi 00100, Kenya; richard2powell@yahoo.co.uk

**Keywords:** grief, bereavement, parents, low- and middle-income countries (LMICs)

## Abstract

While great strides have been made in improving childhood mortality, millions of children die each year with significant health-related suffering. More than 98% of these children live in low- and middle-income countries (LMICs). Efforts have been made to increase access to pediatric palliative care (PPC) services to address this suffering in LMICs through policy measures, educational initiatives, and access to essential medicines. However, a core component of high-quality PPC that has been relatively neglected in LMICs is grief and bereavement support for parents after the death of their child. This paper reviews the current literature on parental grief and bereavement in LMICs. This includes describing bereavement research in high-income countries (HICs), including its definition, adverse effect upon parents, and supportive interventions, followed by a review of the literature on health-related grief and bereavement in LMICs, specifically around: perinatal death, infant mortality, infectious disease, interventions used, and perceived need. More research is needed in grief and bereavement of parents in LMICs to provide them with the support they deserve within their specific cultural, social, and religious context. Additionally, these efforts in LMICs will help advance the field of parental grief and bereavement research as a whole.

## 1. Introduction

Individuals under the age of 20 comprise about 35% of the world’s population, rising to 40% in the least developed nations [1]. While great strides have been made in addressing infant and childhood illnesses globally, as many as 2.5 million children die each year with serious health-related suffering [2]. More than 98% of these children live in low- and middle-income countries (LMICs) [2], with many presenting late to care with significant disease burdens and a limited role for curative options, which compounds their suffering. Therefore, there is a critical need for pediatric palliative care (PPC) interventions in LMICs. Grief and bereavement support to parents whose child has died is a key component of high-quality PPC.

The World Health Organization (WHO) defines palliative care as the prevention and relief of suffering of patients and their families facing the problems of life-threatening illness. These problems include the physical, psychological, social, and spiritual suffering of both the patients and their family members [3]. While the WHO definition applies to the entire lifespan, palliative care for children has unique features requiring special attention and training. These include developmental stages and stage-appropriate communication, weight-based medication dosing, and unique clinical conditions with uncertain prognostication, among others [3,4,5]. In 2014, the World Health Assembly (WHA) published resolution WHA67.19, which committed to strengthening palliative care as a component of comprehensive care throughout the life course, emphasizing that access to palliative care for children is an “ethical responsibility of health systems” [6]. 

Despite the need for PPC, such services in LMICs remain variable at best. A 2011 systematic review demonstrated that 65.6% (*n* = 126) of countries had no PPC services available, 18.8% (*n* = 36) had capacity building activities, and only 5.7% had activities that reached mainstream providers (*n* = 11) [7]. The *Global Atlas of Palliative Care at the End of Life* has also demonstrated that lower income countries have the greatest number of children with palliative care needs at the end of life, but the least amount of access to such interventions [8].

Since 2011, efforts have been made to improve PPC globally through policy measures, educational initiatives, and increased access to essential medicines [3,9,10,11,12]. These efforts have helped increase access to palliative care for pediatric patients around the world. However, a core component of PPC that has been relatively neglected in LMICs is grief and bereavement support for parents after the death of their child. A 2010 survey evaluating access to PPC for children with cancer in economically diverse regions around the world found that the availability of bereavement care was the least available PPC service in resource-limited settings [13]. There was also a significant difference in the perception of bereavement care as quality PPC between low-income countries (LICs) and high-income countries (HICs). 

This manuscript aims to address this limitation by reviewing the current literature on parental grief and bereavement in LMICs. This includes evaluating the current state of bereavement research in HICs, including its definition, adverse effect upon parents, and supportive interventions, before reviewing the literature on health-related grief and bereavement in LMICs, specifically around: perinatal death, infant mortality, infectious disease, interventions used, and provider comfort and perceived need. This review does not include all potential causes of parental grief and bereavement. These include ambiguous loss, such as displacement and disappearances secondary to war or famine, which may also be powerful causes of grief and bereavement in parents but are beyond the scope of this review [14]. For the purposes of this manuscript, we will focus on health-related mortality of children and the subsequent impact that has on parents.

## 2. Methods

A narrative literature review was performed to provide a diverse perspective of parental bereavement in LMICs instead of attempting to answer a specific clinical question [15]. A search of the literature was performed in December 2019 and included any paper published in English found in PubMed and the Cumulative Index of Nursing and Allied Health Literature (CINAHL) (any publication date through December 2019) with medical subject headings (MeSH) terms including bereavement, grief, parents, father, mother, global, Africa, Asia, Middle East, Central America, South America, low-income, middle-income, and LMIC. A hand search of all the retrieved literature was also performed looking at the references of each identified article. In addition to the database search, discussions with experts in the field of PPC in the global setting were performed to identify other articles and publications. Articles were included if they specifically evaluated the bereavement experience of parents after the death of a child in a LMIC. A total of 183 papers were assessed and 11 describing parental grief and bereavement in LMICs were included in this manuscript.

## 3. Current State of Bereavement Research in High-Income Countries

### 3.1. Definitions

While no universal definition for bereavement currently exists in the literature, it is important to differentiate bereavement from other related terms often associated with it. Bereavement is the objective situation one experiences after the death of an important person in their life [16]. Mourning, a related term, refers to the public display of grief [16]. Grief itself deals with the emotional/affective process of reacting to the loss of a loved one through death [16]. It is important to note that these definitions are specific to the English language and primarily Western cultural context, and different languages and cultures may combine these terms into one word or phrase. For the purposes of this paper, however, we will use these definitions for the description consistency and comparison between studies. 

Researchers have proposed and described a variety of different models for the expression of grief and have attempted to clarify the difference between “normal” and “complicated” grief, as well as its processes [16,17,18]. It is beyond the scope of this review to describe grief in all of its forms but it is important to note that, in our clinical experience, the grief reaction to the loss of a loved one is a uniquely individual experience based on the person’s relationship to the patient, previous experiences, and their specific cultural context. This is especially true when the bereavement is due to the death of a child. 

### 3.2. Parental adverse effects

The death of a child, expected or not, is one of the most difficult experiences a parent faces. Bereavement after the loss of a child has been found to elicit a more intense grief reaction than to losing a spouse or parent [19]. Additionally, loss of a child can have significantly adverse effects on a parent’s physical and psychological well-being. Parents who have lost a child are at risk of having long-term detrimental effects, such as depression or anxiety [20,21], as well as increased psychiatric admissions [22]. Parental bereavement can also have an impact on the social framework of families, with some studies showing evidence of increased marital discord and divorce [23,24]. Other studies, however, have not replicated these results and have found continued marital stability even after the death of a child [25,26]. Along with the psychological impact, evidence exists that parental bereavement can have negative impacts on the physical health of the parents, including poorer health-related quality of life [27,28], increased risk of cardiac morbidity [29], and even increased mortality [30]. 

### 3.3. Supportive interventions

Due to the psychological, social, and physical health impacts of parental bereavement, several different interventions have been studied to assist bereaved individuals. In 2004, a systematic review evaluated bereavement care interventions from 74 eligible studies. The interventions included pharmacotherapy for bereavement-related depression, support groups and counselling, psychotherapy (cognitive-behavioral, psychodynamic, psychoanalytical, and interpersonal therapies), and systems-oriented interventions [31]. Other than pharmacotherapy for bereavement-related depression, there were insufficient rigorous evidence-based recommendations regarding the treatment and support of bereaved individuals. 

A follow-up systematic review in 2015 by Endo et al. looked specifically at interventions for bereaved parents and identified nine eligible articles studying the impact of four different types of interventions, including support groups, counseling, psychotherapy, and crisis intervention [32]. Once again, they concluded that the lack of rigorous evidence precluded any formal recommendation about interventions for bereaved parents. Most recently, in 2019, another systematic review by Ainscough et al. also looked at the effectiveness of bereavement support for parents of infants and children [33]. Of the 24,556 records assessed, 9 articles met the inclusion criteria. Of those 9, 5 were randomized control trials with 23 different outcomes measures. Only 3 of the 9 studies reported a significant difference between the control group and the experimental group, despite the wide range of outcomes measured. Like the other systematic reviews, Ainscough et al. emphasized the need for more robust study methodologies and an agreement on core outcomes to assist in further research on interventions for bereaved parents. 

### 3.4. Health-Related Grief and Bereavement in LMICs

It should be noted that the research on the impact of grief and bereavement on parents’ psychological and physical health, along with the interventions for parental grief and bereavement, are from publications predominantly originating from within North America and Western Europe. This current literature may not be sufficient or inclusive of people in different cultural and socioeconomic contexts around the world. In one comparison between bereavement counsellors’ experiences in a Western country (Northern Ireland) and an African nation (Uganda), there were significant differences in cultures and the lived grief experience [34]. In particular, they found that the individualistic culture and approach to bereavement support in Northern Ireland contrasted with the more communal, collectivist experience in Uganda. Therefore, it is vital to better understand how the varied cultural, religious, and spiritual contexts of bereaved families impact their ability to attribute meaning after the death of their child [35]. 

The evidence in the current literature about parental bereavement in underserved communities around the world is limited. From the studies we did identify, several different causes of death in children were investigated including perinatal death, infant mortality, and infectious disease among others. Of note, there were no studies we could find that assessed parental bereavement in LMICs after the death of a child due to cancer. A summary of the papers is found in Table 1 and a map of the low- and middle-income countries where these studies were performed is found in Figure 1. 

### 3.5. Perinatal Death 

The most robust research in parental bereavement in LMICs is evaluating the impact of perinatal death on parents. A systematic review of parents’ and professionals’ experiences after stillbirth in LMICs was published in 2017 by Shakespeare et al. and evaluated 34 studies from 17 different countries [36]. Over half of the studies included were qualitative in nature. They identified 13 themes which were consistent between the different studies and a frequency effect size was calculated to reflect in how many studies each particular theme appeared. Those themes which appeared in over 50% of the studies included: 

Positive community support, as opposed to stigmatization and blame, can improve the bereavement experience.

Awareness of and support for appropriate coping mechanisms can assist grieving.

Women’s experience of grief, which has multiple manifestations, often goes unrecognized by the healthcare community and wider society.

Access to timely and culturally appropriate psychological support is valued.

Addressing health system barriers is important for provision of respectful care.

Women want information, advice, and individualized discussions about future pregnancies. 

One other important theme was that bereaved women may experience devaluation and stigmatization as a result of cultural practices and beliefs, where there was 47% frequency effect size. A powerful message from this review was how culturally heterogenous the healthcare provision was among the different LMICs, even within the same country. Even with such cultural differences and varied societal contexts, a key theme throughout the different studies evaluated was the negative experience of women who experience stillbirth, and the blame, stigma, and lack of support from family, their community, and healthcare workers. Several studies identified the cultural stigma about perinatal death and repressed opportunities to grieve [38,47,48]. 

While most studies looked at the experience of perinatal death for mothers, two studies have looked at the experience of bereavement in fathers after stillbirth. The first interviewed 15 fathers from northeastern Columbia whose child died either as a stillbirth or before 7 days of age [37]. The authors found that the fathers often suffered alone as they focus on the suffering of their partner. The men also felt neglected and forgotten by the hospital staff, with several fathers not allowed in the room to support their partners or to have meaningful moments with their child. This led to significant distress and the authors concluded that healthcare professionals should seek opportunities to support fathers in their bereavement after perinatal death. 

The second article interviewed five key informants to discuss the impact of stillbirth on men in India [38]. Several themes emerged around the man’s rights over his partner’s to medical and reproductive decision-making. After a stillbirth, mothers were discouraged from expressing their grief and pushed to conceive again. If conception was unsuccessful or if the child was a daughter and not a son, this was still considered undesirable and the husband would consider finding another wife. Quantitative data demonstrated that men with a history of stillbirths had greater anxiety and depression and perceived less social support. They had more progressive views towards women than men without stillbirth experience but fathers with a history of stillbirths were also more likely to be emotionally or physically abusive. The authors concluded that bereavement resources should consider the father’s attitudes and behaviors towards reproductive decision making. 

### 3.6. Infant Mortality

While there were over 30 studies on perinatal mortality and its impact on parental bereavement, there is significantly less in the literature with regards to other causes of parental grief. Three studies looked at general infant mortality and its impact on parental grief. Meyer et al. interviewed eight mothers in Ghana about the experience of the death of an infant. The major themes included little discussion about the etiology of their child’s death and a discouragement in discussing or thinking about their child after they had died [39]. Seven out of the eight stated that they held no memorial service and that the culture of silence and not discussing the child, or their death, was prevalent. Six out of the eight mothers were explicitly told not to discuss their child and to forget their loss as quickly as possible, and only two of the eight mothers felt that this avoidance was beneficial in their grief process. The mothers did state that those factors that helped them in their coping and bereavement included their faith and their other children. 

Fouts and Silverman evaluated the interactions between mothers and their children after the death of another child within Bofi foragers in the Central African Republic (CAR) [40]. They evaluated the theory that increased environmental risk and high infant mortality impacted parents’ subsequent parenting strategies. Bofi foragers live in and around the Congo River Basin within the CAR and have high birth rates and infant mortality rates. They followed 30 mothers who had given birth to at least two children. Twenty-six of the thirty mothers had experienced the loss of a child. What they found was that those children whose mother had lost two children were significantly more likely to be held and have significant physical contact than those children whose mothers had lost zero to one child, and they were also held more as compared to those children whose mothers had lost three or more children. They theorized that while losing multiple children increased mothers’ interactions with their children, there comes a “saturation point” in which parents’ investment was perceived as not increasing a child’s probability of survival. 

Finally, Goldstein et al. evaluated grief responses to sudden infant death in different contexts, including mothers in South Africa, a Native American reservation in Sioux Falls, South Dakota, United States of America (USA), as well as in higher income locations in the USA, United Kingdom, New Zealand, Australia, and the Netherlands [41]. In comparing the different groups, they found high rates of prolonged grief disorder that were not significantly different despite the extremely varied and diverse cultural contexts. They concluded that while culture may impact grief expression, there is an ingrained response to child loss that is maintained regardless of cultural circumstances. 

### 3.7. Infectious Disease

In most developed countries, the emphasis on end-of-life and bereavement care is focused on non-communicable diseases, specifically cancer. However, in LMICs, a high proportion of childhood mortality is due to infectious diseases. A systematic review evaluating end-of-life care in Sub-Saharan Africa identified that of the 51 articles included, 38 (74.5%) dealt with Human Immunodeficiency Virus/Acquired Immunodeficiency Syndrome (HIV/AIDS) [49]. One study showed the importance of the community in helping others who had experienced loss to HIV/AIDS [50]. However, the Gysels et al. systematic review also identified three studies that showed how bereavement and grief were complicated after HIV/AIDS due to the social stigma and silence associated with that diagnosis [49]. However, all of these studies were undertaken among adults generally, so little is known about the specific experience of parental bereavement. 

One study that evaluated the experiences of 10 mothers who had lost young children to HIV/AIDS, and 12 key informants, was undertaken in KwaZulu-Natal, South Africa [42]. The authors identified several stresses, including remaining silent about their grief due to the fear and persistent stigma about HIV/AIDS. These mothers also felt guilt for feeling like they were the cause of their child’s sickness. They also struggled to find time to grieve as they faced the challenges of extreme poverty. Both the mothers and key informants felt that talking about their grief and bereavement experience would be beneficial, but the mothers felt stressed by their financial and social challenges and the key informants also identified a need for better training and support for mental health services to help these mothers in their bereavement journey. 

### 3.8. Interventions

The majority of the literature on grief and bereavement in LMICs are qualitative studies with structured or semi-structured interviews of parents or key informants describing their experience. Very few studies assessed interventions for bereaved parents. One study, performed in Turkey, created a grief support program for parents whose child had died. The couples were divided into a control (38 couples) and intervention group (39 couples) [43]. The intervention included brochures and other literature about grief and bereavement, as well as interactions with a researcher, and included a variety of home visits and phone calls at various time points before the child’s death, at the time of death, and after the child’s death. They found that there was no difference between the control and intervention groups at 3 and 6 months, but at 12 months there was, but it was not statistically significant. The authors did note that the cultural and religious rituals, that had been assumed to help in the grieving process, did not seem to impact the grief of parents after the loss of a child. Multiple other studies described the fact that the cultural rituals actually hindered rather than supplemented the grief and bereavement process in parents [37,47,48]. 

Another intervention described in the literature is the nine-cell bereavement tool [44,45]. The goal of this tool is to assist in describing the bereavement process in a culturally sensitive manner. It consists of horizontal cells showing reactions at the time of death, after several months, and after several years, with vertical cells describing the personal feelings of the parent, what they outwardly expressed, and what is allowed/expected in the culture. The tool has been implemented in several settings including India, Kenya, Zimbabwe, and South Africa, as a method to train community health workers, palliative care workers, and home-based care providers. In follow-up interviews after the training, participants stated that they learned the impact culture has in mediating grief reactions and that the bereaved may behave normally but that this does not mean that they are not feeling pain [45]. Additionally, they learned that it is important for the bereaved to discuss their grief, and to find questions to encourage them to do so. 

### 3.9. Provider Comfort and Perceived Need 

While the majority of the studies on grief and bereavement focused on parents in LMICs, one study looked at healthcare providers and their perception of bereavement care. Chan et al. assessed 573 nurses working in labor and delivery, newborn nursery, and neonatal intensive care units within Singapore, Hong Kong, and Jinan, China, about their comfort addressing topics of bereavement in the perinatal population [46]. They found that more senior nurses, those who had previous encounters with bereaved parents, and those that had received training on palliative care and bereavement, felt more comfortable working with bereaved families. Most of the nurses in the study agreed or strongly agreed that training in bereavement care is important. This was seen in several other studies that interviewed key informants and healthcare providers as they felt that further training and access to bereavement resources is important for families grieving the loss of a child [42,45]. 

## 4. Conclusions

The bond between parent and child is one of the most powerful and significant relationships between human beings. Consequently, the death of a child is one of the most difficult things a parent may ever face in their life as their identity of provider and protector is lost. There is now increasing evidence on the impact this loss has on parents psychologically and physically, but the majority of this research is within HICs. Therefore, grief and bereavement care for parents after the death of a child is a fundamental component of high-quality PPC and should be included in efforts of early integration of palliative care and universal health coverage promotion.

More research is needed in grief and bereavement of parents in LMICs to help provide them with the support they deserve within their specific cultural, social, and religious context. In particular, it is vital to better understand grief and bereavement of fathers along with how gender stigma impacts the loss of a child and ways to overcome this stigma. This research also includes the need to develop core outcome measures to evaluate bereavement interventions as well as more methodologically robust study designs to test these interventions in a culturally sensitive manner. It will also be important to investigate the views of healthcare providers and identify perceived barriers to providing quality bereavement care specific to the context and community where they work and identify potential solutions to overcome these barriers. In addition, more education is needed on parental grief and bereavement not just for healthcare providers and bereaved families, but also for the general public as well to overcome barriers to bereavement care. 

It is critical to understand the rich diversity of bereavement practices around the world and the need to respect the unique cultural differences. With that, it is also a powerful message that in the literature there exists similarities between parental bereavement despite differences in culture and background. By better understanding the bereavement experience of parents in LMICs and the interventions that can best help these parents, we can ensure that they receive the high-quality care that they deserve through their most challenging moments. Additionally, these efforts in LMICs will help advance the field of parental grief and bereavement research as a whole. 

## Figures and Tables

**Figure 1 children-07-00039-f001:**
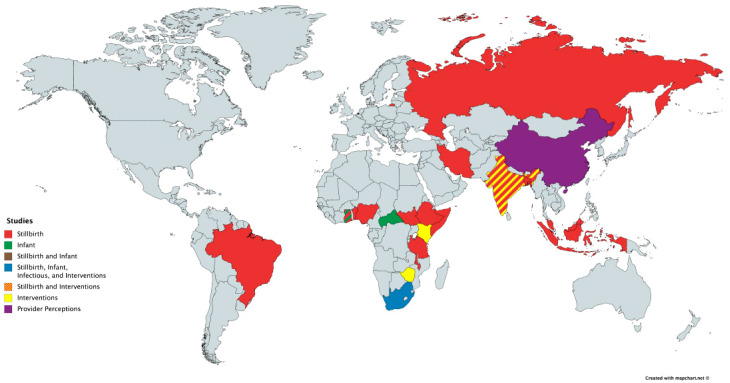
World map with study location and type.

**Table 1 children-07-00039-t001:** Papers evaluating parental grief and bereavement in low- and middle-income countries.

Paper	Year	Type of Paper	Countries Evaluated	Key Findings
**Stillbirth**				
Shakespeare et al. [36]	2019	Systematic Review	Bangladesh, Benin, Brazil, China, Ethiopia, Ghana, India, Indonesia, Iran, Malawi, Malaysia, Nigeria, Russia, Somalia, South Africa, Tanzania, Uganda	Common themes from the review include: positive community support, as opposed to stigmatization and blame, can improve the bereavement experience. Women’s experience of grief is often unrecognized by the healthcare community and wider society and that access to timely and culturally appropriate psychological support is valued.
Lizcano Pabon et al. [37]	2019	Qualitative Study	Columbia	Fathers often suffer alone as they focus on the suffering of their partner. They felt neglected and forgotten by the hospital staff and were not allowed in the room to support their partners or to have meaningful moments with their child.
Roberts et al. [38]	2017	Mixed Methods Study	India	Women were discouraged from grieving a stillbirth and pushed to conceive again. Men with a history of stillbirths had greater anxiety and depression and perceived less social support and were also more likely to be emotionally or physically abusive.
**Infant**				
Meyer et al. [39]	2016	Qualitative Study	Ghana	Mother’s were discouraged in discussing or thinking about their child after they had died. Few mothers felt that this avoidance was beneficial in their grief process.
Fouts and Silverman [40]	2015	Observational Study	Central African Republic	Those children whose mother had lost 2 children were significantly more likely to be held and have significant physical contact than those children whose mothers had lost 0 to 1 child and they were also held more as compared to those children whose mothers had lost 3 or more children.
Goldstein et al. [41]	2018	Cross-Sectional Survey	South Africa	High rates of prolonged grief disorder were not significantly different despite extremely varied and diverse cultural contexts. While culture may impact grief expression, there is an ingrained response to child loss that is maintained regardless of cultural circumstances.
**Infectious**				
Demmer et al. [42]	2010	Qualitative Study	South Africa	Several stressors impact a mother’s grief including remaining silent about their grief due to the fear and persistent stigma about Human Immunodeficiency Virus/Acquired Immunodeficiency Syndrome (HIV/AIDS). The mothers also felt guilt for feeling like they were the cause of their child’s sickness. They struggled to find time to grieve as they faced the challenges of extreme poverty.
**Interventions**				
Yildiz et al. [43]	2017	Intervention study with pre- and post-test control group	Turkey	A grief support program was created for parents whose child had died. The intervention included brochures and other literature about grief and bereavement as well as interactions with a researcher at various time points. There was no difference between the control and intervention groups at 3 and 6 months. At 12 months, there was a difference that was not statistically significant.
Hunt 2002, Hunt et al. 2007 [44,45]	2002, 2007	Intervention tool development and qualitative feedback	India, Kenya, South Africa, Zimbabwe	Participants trained with this tool learned the impact that culture has in determining grief reactions and that the bereaved may behave normally but that this does not mean they are not feeling pain.
**Provider Perceptions**				
Chan et al. [46]	2010	Cross-Sectional Survey	China, Singapore	More senior nurses, those who had previous encounters with bereaved parents, and those that had received training on palliative care and bereavement, felt more comfortable working with bereaved families. Most of the nurses in the study agreed or strongly agreed that training in bereavement care is important.

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
