# Peer review of "Grief and Bereavement in Parents After the Death of a Child in Low- and Middle-Income Countries"

_children, 2020, doi:10.3390/children7050039_

Round 1

Reviewer 1 Report

I generally found this to be a solid paper. The focus on grief and bereavement among parents after the death of a child in Low-and Middle-Income countries is a critical issue and one that has been given little attention. 

The only suggestion I have is for the Conclusion to consider the need:

  • for more research on fathers, 
  • for more understanding of how gender stigma affects this trauma and ways to assist/resolve 
  • for more education - not just for healthcare providers and affected families ...but for the general public as well.

The only need for clarification relates to your discussion of reference 41 (line 211-215). Several points seem to conflict.

Reviewer 2 Report

Overall, I think this was a thoroughly researched and well written paper that will further contribute to the scientific community. The largest strength of the paper is recognizing how little we actually know about the topic and how much additional work and research is still required.

Prior to publication, it would be helpful to review Table 1. I would recommend really trying to hone in on the major point of each article. You have the opportunity to further expound upon the themes in the sections following this table, and you do. Much of what followed the table felt like a verbatim repetition of the information in the table. Try to limit the key findings to just that- the key findings. You can further defend and explain these key findings in the body of the manuscript and not in the table. Additionally, consider adding a space or some other differentiation the key findings of the individual papers. There were certain points where it was hard to determine where the key points of one paper ended and the key points of the next paper began. Specifically look at the key points between Shakespeare et al. and Lizcano Pabon et al. This should be clearer to your audience.

Repeatedly throughout the paper you cite the same paper several times within the same paragraph. You introduce the paper in the first sentence and cite it here. Then you summarize the results of this paper and only this paper without citing other papers for the duration of the paragraph. In these instances I do not believe that you need to repeatedly provide the citation following every sentence as it is assumed you are still discussing the originally cited paper.

Line 239- there is an extra end parenthesis that does not need to be there.

Line 286- it appears that you are citing the same paper but using multiple means of citation. You should remove the “(Chan et al. 2010. Eval and Health Professions)” reference.
